# *In vitro* Comparative Study of Fibroblastic Behaviour on Polymethacrylate (PMMA) and Lithium Disilicate Polymer Surfaces

**DOI:** 10.3390/polym11040744

**Published:** 2019-04-25

**Authors:** Cristina Herráez-Galindo, María Rizo-Gorrita, Irene Luna-Oliva, María-Ángeles Serrera-Figallo, Raquel Castillo-Oyagüe, Daniel Torres-Lagares

**Affiliations:** 1Department of Oral Surgery, College of Dentistry, Seville University, 41009 Seville, Spain; crisnach.15@gmail.com (C.H.-G.); marrizgor@alum.us.es (M.R.-G.); irene_ire337@hotmail.com (I.L.-O.); maserrera@us.es (M.-Á.S.-F.); 2Department of Dental Protheses, College of Dentistry, Complutense University of Madrid, 28040 Madrid, Spain; raquel.castillo@odon.ucm.es

**Keywords:** polymethyl methacrylate (PMMA), cytomorphometry, lithium silicate, gingival fibroblast, computer-aided design/computer-aided manufacturing (CAD/CAM), early cell proliferation

## Abstract

Polymethyl methacrylate (PMMA) and lithium disilicate are widely used materials in the dental field. PMMA is mainly used for the manufacture of removable prostheses; however, with the incorporation of CAD-CAM technology, new applications have been introduced for this material, including as a provisional implant attachment. Lithium disilicate is considered the gold standard for definitive attachment material. On the other hand, PMMA has begun to be used in clinics as a provisional attachment until the placement of a definitive one occurs. Although there are clinical studies regarding its use, there are few studies on cell reorganization around this type of material. This is why we carried out an *in vitro* comparative study using discs of both materials in which human gingival fibroblasts (HGFs) were cultured. After processing them, we analyzed various cellular parameters (cell count, cytoskeleton length, core size and coverage area). We analyzed the surface of the discs together with their composition. The results obtained were mostly not statistically significant, which shows that the qualities of PMMA make it a suitable material as an implant attachment.

## 1. Introduction

Since the first treatment developed with the aid of a CAD/CAM system (Computer-Aided Design/Computer-Aided Manufacturing) was created in 1985, this technology has continued to evolve, giving way to the new era of digital dentistry. These changes have prompted the production of a new range of next-generation dental materials that can be milled using this system (e.g., PMMA and lithium disilicate) [1,2,3,4,5,6,7,8,9,10,11,12].

Ceramics have been used for a century in dentistry [2,13,14,15]. This group includes lithium disilicate ceramics (Li_2_Si_2_O_5_) that were introduced in dentistry in 1998. In 2006, the IPS e.max CAD (Ivoclar Vivadent, Schaan, Liechtenstein) was introduced to the market, on which our study was carried out [4,6,7,9,14,15]. The blocks of this material are supplied in a pre-crystallized blue state. This ceramic contains metasilicate and lithium disilicate cores in its interior, which has excellent mechanical properties (flexural strength of 130 ± 30 MPa). In this state, the block can be easily milled, and then the structure is crystallized in a ceramic oven at 850 °C under vacuum for 20–25 min. During this process, the metasilicates dissolve, leaving lithium disilicate crystals that will be glazed, thus producing the desired color change. The material can then achieve a strength of up to 360 MPa according to the manufacturer [4,6,7,9,14,15]. This monolithic restoration material is not only suitable for the manufacture of implant crowns but also for the manufacture of inlays, onlays, crowns and veneers [4]. 

The other material we studied is polymethyl methacrylate (PMMA), widely used since the 1930s for the manufacture of removable prostheses, orthodontic appliances and splints. This is a synthetic polymer that provides strength, color stability and ease of repair, some of the essential qualities required for provisional material. Biomaterials such as PMMA may be used to maintain permanent or temporary contact with the body in order to maintain cells and tissues [16,17,18]. In 2005, PMMA-based polymer, Vita CAD-Temp material was introduced (VITA Zahnfabrik, Bad Säckingen, Germany), an improved polymer that uses PMMA in its composition and enables manufacture using a CAD/CAM system. In addition to its well-known specifications, it is also used as a material for long-term provisional implant crowns due to its high modulus of elasticity and flexural strength [16,19]. Its main disadvantages are shrinkage during curing, exothermic reaction during polymerization and release of monomers into the medium. All of these are diminished when produced by a CAD/CAM system with controlled polymerization under optimum pressure and temperature (the manufacturer even ensures free monomer absence). This system also reduces laboratory costs and speeds up the process [4,7,19].

When selecting an implant material, it is important to take its biocompatibility into account, as well as its cellular behavior around these surfaces. After the osseointegration period of the implants, transgingival abutments are inserted to reshape the gingival tissues. The healing of both the bone tissue and soft tissues surrounding the various surfaces of the implants is essential for clinical success [10,20,21,22]. These prostheses connect the internal environment of the soft tissue with the oral environment, which is required to create a hermetic structure where cell proliferation is generated that prevents the passage of bacteria to the peri-implant zone [8,9,10,22,23]. After implant placement and insertion of implant abutments, the marginal gingiva is often irritated and has to regenerate. First of all, the wound healing starts with the inflammatory phase, which is followed by a proliferative phase, during which fibroblasts migrate, accumulate into the wound and produce a collagen matrix. This phase ends with the keratinocytes migration. These two cell types create the barrier. The cells’ adhesion depends on various components such as lipids, carbohidrats and bound proteins [6,24,25,26,27,28,29]. 

As far as we know, to date, no study has been published in which these two materials (Vita CAD-Temp and IPS e.max CAD) are analyzed related to the proliferation, morphology and spread of fibroblasts. That is why we focus on the comparison of these materials’ surfaces, as well as the cellular response thereon.

Our null hypothesis is that both materials have the same behavior at the level of proliferation, cell morphology or spreading. Lithium disilicate ceramic is considered a gold standard for definitive restorations on implants and natural teeth. PMMA, on the other hand, is widely known as a long-term provisional material.

## 2. Materials and Methods 

### 2.1. Preparation of Samples

We exposed the two materials under study, IPS e.max CAD (Ivoclar Vivadent, Schaan, Liechtenstein) and Vita CAD-Temp (VITA Zahnfabrik, Bad Säckingen, Germany), in accordance with manufacturer reference data (Table 1).

Using blocks of both restoration materials (definitive (IPS e.max CAD) and long-term provisional components (Vita CAD-Temp)), discs were milled using the Straumann^®^ CARES^®^ M Series CAD/CAM system (Straumann Group, Basel, Switzerland; tolerance value ± 3 µm). The laboratory used Software CARES^®^ Visual (Straumann Group, Basel, Switzerland) to design the disc and the Program 3 in the milling process (26 °C during 9 h 40 min). This CAD/CAM system works at 250 W, 60,000 rpm, irrigated with water and cutting fluid and with a precision < 10 μm.

We prepared the PMMA through milling, whereas the lithium disilicate must be sintered after milling, as previously mentioned. Discs comprised an 8-mm diameter and 2-mm height, and these values were obtained after milling (PMMA) and milling and sinterization (lithium disilicate). 

Once the disks were readied, we prepared them through sterilization with short-wavelength (200–280 nm) UVC rays with exposure for 30 min on each side in a laminar flow hood. Next, they were placed onto a sterile petri dish prior to beginning the experiment.

### 2.2. Surface Analysis

#### 2.2.1. Scanning Electron Microscopy

We carried out FESEM (Field Emission Scanning Electron Microscopy) using a FEI TENEO scanning electron microscope (FESEM; Thermo Scientific Inc., Waltham, MA, USA) at a magnification of 200×.

#### 2.2.2. Profilometry

We used a Sensofar S NEOX interferometric confocal microscope (Sensofar Medical, Terrassa, Spain) to obtain the roughness using SensoMAP Premium 7.4 software (Sensofar Medical, Terrassa, Spain). Then we used epifluorescent lenses 20× magnification at a length of 4.50 mm and a green optical resolution of 0.32 μm. Five random measurements were made from different locations for each material, with a pre-set dimension of 0.87 × 0.66 mm^2^ and a compensation of 250 μm. These measurements are based on ISO 25178 with regards to the geometric specifications of the product. The quantitative roughness parameter used was the arithmetic mean of the standard deviation of 3D roughness (Sa).

#### 2.2.3. Composition

Prior to beginning the experiment, we analyzed the surface composition of the discs using energy dispersive spectroscopy (EDS) analysis with a FESEM (Thermo Fischer Scientific Inc., Waltham, MA, USA) that includes Schotkky diode field emission and an EDAX AMETEK SDD (AMETEK, Leicester, UK). To process the images that were obtained, we used EDAX TEAM software version 4.4.1 (AMETEK, Leicester, UK). We analyzed an area of 130 μm in 200 s to obtain results expressed in mass percent composition (wt %).

### 2.3. Cell Culture

Human gingival fibroblasts (HGFs; Lonza, Basel, Switzerland) were incubated in T75 flasks and placed in an incubator (Nuaire US Autoflow, CO_2_ Water-Jacketed Incubator) at 5% CO_2_, 95% air and 37 °C.

We then added a ready-made culture medium, Dulbecco’s Modified Eagle Medium (DMEM, Biowest, Nuailé, France), supplemented with 10% bovine serum (FBS, Biowest, Nuaillé, France) and 1% glutamine-penicillin-streptomycin (Biowest, Nuaillé, France).

To monitor initial adhesion, growth and cell expansion between each pass, we verified the state of the cells in the T75 flasks using an Olympus CKX41SF2 microscope (Olympus, Shinjuku-ku, Tokyo, Japan). We always placed the cells in the discs between the third and tenth pass, as recommended by the manufacturer. 

We placed the specimens of both materials onto a sterile petri dish to cultivate fibroblasts on them at a concentration of 1 × 10^3^ cells in 40 μL of growth medium in each dish. After a few hours, we checked the initial cell adhesion to the disc surface using an Olympus CKX41SF2 microscope (Olympus, Shinjuku-ku, Tokyo, Japan). Next, we placed the medium onto the petri dish using a pipette until it covered the surface of the discs. We incubated the cells for 24 h prior to carrying out immunohistochemical staining.

### 2.4. Immunohistochemical Staining

Next, using a pipette, we carefully removed the medium and then washed the surface twice with 500 μL of DPBS (AppliChem GmbH, Darmstadt, Germany). Subsequently, we fixed cells with paraformaldehyde in DPBS (AppliChem GmbH, Darmstadt, Germany) for 10 min and then aspirated and washed them again with DPBS prior to permeabilization with 15 μL of 0.1% Triton X-100 (Sigma, Saint Louis, MO, USA) at 4 °C for 5 min.

Afterwards, we aspirated the Triton and then washed the specimens twice with DPBS once again. To block this phase, 1% bovine serum albumin (BSA, Biowest, Nuaillé, France) was applied for 20 min. Later, we removed the excess with a pipette. Subsequently, we carried out immunohistochemical staining with 15 μL of Fluorescent Phalloidin 488 (Cytoskeleton, Inc., Denver, CO, USA) on each disc. We used this type of stain due to its affinity for F-actin, which is found in the filaments of the cellular cytoskeleton.

We stained the fibroblast nucleus with 15 μL of 4’,6’-diamino-2-phenylindole (DAPI), which was a component of the VECTASHIELD Mounting Medium (Vector Laboratories Inc., Burlingame, CA, USA). We left the discs stained with phalloidin in the dark for 30 min before being washed again with DPBS, and we applied DAPI to the surface.

We maintained the discs at 4 °C and observed them using a Zeiss LSM 7 DUO electron microscope (Carl Zeiss, Jena, Germany) after 24 h using an X-Cite 120 PC epifluorescence unit (Excelitas Technologies, Waltham, MA, USA).

### 2.5. Microscope Viewing 

To obtain microscopic images, we used fluorescein isothiocyanate (FITC) at 488 nm to capture the fluorescence emitted when passing through the cytoskeleton (collected in a filter that captured a wavelength of 535 nm). To view the nuclei, we used a DAPI excitation filter at 355 nm, capturing the fluorescence emitted at 458 nm through the filter. We carried out the entire process in accordance with the parameters recommended by the manufacturer.

The software we incorporated into the microscope was the Zen Lite 2012 (Carl Zeiss, Jena, Germany), through which the images could be viewed at different objectives 20×/0.8 and 40×/1.30 with immersion oil and 63×/1.40 with oil. We captured the images in five regions of interest (ROI) for each disc: north-west, north-east, center, south-west and south-east. No sample was exposed to the laser any longer than 5 min so as to prevent burning.

### 2.6. Image Processing

We processed the obtained images using ImageJ v1.50e software (Wayne Rasband, National Institutes of Health, Bethesda, MD, USA). We saved these images in .tiff format to be viewed using this program, with 2048 × 2048 pixel dimensions. Once the images were opened and calibrated, we changed them to 8-bit. Those obtained using a 20× objective served to analyze the number of cells covering the surface, whereas those obtained at 40× and 63× served to analyze the morphology and to perform measurements of cell size.

These data were obtained using various plug-ins incorporated into the program. Additionally, we passed the image through three RGB color channels (red, green and blue), maintaining the blue image, which showed the nuclei limits [30].

We also used the Watershed command to automatically separate the nuclei of each cell. The next step was for us to analyze each particle, including smaller particle sizes of 5 μm^2^.

Finally, to obtain cell length measurements, we used three random cells in each of the images acquired at 40× magnification, drawing a line along the main axis, so as to obtain the results in μm.

### 2.7. Cell Parameters Analysed

We analyzed certain cell variables, such as the number (proliferation), average nuclei size (core size), percentage of area occupied by nuclei in relation to the total image (percentage of nuclei coating), circularity and average length of cytoskeleton axes (cell length).

Cell parameters obtained may be observed in the results section, based on the SEM images, and we always captured said data as a standard deviation for each of the variables.

The value of circularity oscillates between 0 and 1 [10], with values closest to 0 being recognized as elongated sizes and those closest to 1 as circular, based on the formula to which the perimeter corresponds with double π multiplied by two radii (i.e., cell diameter).

### 2.8. Statistical Analysis

To compare the results of the two surfaces studied, we used IBM SPSS 24.0 statistical software (International Business Machines Corp, New York, NY, USA). Next, we used the Kolmogorov–Smirnov test to verify that the results were within normal parameters. For variables with normal distribution, we used Student’s t-distribution and, in the case of non-parametric variables, the Mann–Whitney U test, thus establishing a level of significance of *p* < 0.05 [28,29].

## 3. Results

### 3.1. Surface Characterization

#### Surface Topography

When analyzing the images of the samples taken by the scanning electron microscope (SEM), Figure 1 shows the following: On the lithium disilicate (IPS e.max CAD) surface, chips may be seen as a result of milling of the discs and irregularities typical of ceramic sintering. The surface of VITA CAD-Temp, on the other hand, is a more homogeneous surface where concentric grooves may be observed, resulting from the machining of the piece by the manufacturer. 

### 3.2. Profilometry

The average roughness (Sa) from the five values obtained for each material results in IPS e-max being 1.57 ± 0.34 μm and VITA CAD-Temp being 0.38 ± 0.02 μm. There is a greater roughness in the IPS sample due to it being a less homogeneous sample. 

The sample measurement areas and the images obtained from the three-dimensional roughness profile for both materials may be seen in Figure 2.

#### 3.2.1. Analysis of Surface Composition

The results obtained from the EDS analysis of IPS e. max CAD and VITA CAD Temp (Figure 3 and Figure 4) show high crystallization peaks of silicon, oxygen and potassium, whereas the values of potassium, zinc and the other elements located by dispersive spectrometry are much lower. The breakdown of the recognized elements is shown in Table 2. 

Next, we describe the various values obtained in the cell parameters studied (Figure 5).

#### 3.2.2. Cellular Morphology and Anchoring

Figure 6 shows the confocal microscopy images of the fibroblasts, obtained at 40× and 63× magnification, for observation of cell morphology. We used double staining to view the nuclei with DAPI (blue) and the actin filaments of the cytoskeleton with phalloidin (green). 

#### 3.2.3. Cytomorphometry

In relation to nuclear size, the average observed for IPS was 93.48 ± 69.19 µm^2^, and the average was 74.40 ± 32.76 µm^2^ for VITA CAD Temp. According to the Mann-Whitney U test, this difference was not statistically significant (*p* = 0.626).

The mean length of the main cell axis at 40× magnification was 56.12 ± 23.59 μm for IPS and 68.88 ± 31.45 μm for VITA CAD Temp. According to the Mann-Whitney U test, this difference was statistically significant (*p* = 0.038). 

The mean circularity value of the nuclei was 0.57 ± 0.07 for IPS and 0.55 ± 0.14 for VITA CAD Temp. According to the Student’s t-test, this difference was not statistically significant (t_29_ = 0.661, *p* = 0.514).

#### 3.2.4. Cell Count

The average number of cells observed in each image taken at 20× magnification was quantified to assess the effect of surface topography on the cells. In the case of IPS, an average of 115.82 ± 87.68 cells was observed in each image and for VITA CAD Temp, 99.70 ± 91.71 cells. According to the Student’s t-test, this difference was not statistically significant (t_35_ = 0.544, *p* = 0.590).

#### 3.2.5. Covering of the Disc by Nuclei

As for the area covered by cell nuclei, the percentage observed for IPS was 21.26% ± 2.37, and it was 20.57% ± 1.90 in the case of Vita CAD-Temp. According to the Mann-Whitney U test, this difference was not statistically significant (*p* = 0.927). 

## 4. Discussion

The results confirm this study’s null hypothesis, thus finding similar cellular behavior on both surface types, while obtaining very similar figures of the parameters analyzed. 

The Titanium’s blueish-grey color has promoted the use of new aesthetic abutment materials. Many patients, especially those with high esthetic expectations, demand metal-free dentures [6,23], and zirconia and lithium disilicate are the gold standard of aesthetic materials. That’s why we decided to compare one of the gold-standard materials with an interim one (PMMA), and they have a similar cellular behavior according to the results. 

Our aim in this work has been to compare both materials (VITA CAD-Temp and IPS e.max CAD) in relation to fibroblast growth and adhesion to their surface. Implant abutments must have a surface with the ideal characteristics to promote cell adhesion and maintenance of soft tissue health [31].

The clinical importance of the present study is the cellular analysis that we have made to compare two different materials that are already being used as implant abutments in clinical studies. These clinical studies report good results even in the most aesthetic sectors [32,33,34].

No statistically significant differences were found in the majority of the parameters studied (cell count, nuclei size and area covered by cell nuclei), with the exception of cytoskeleton length of the fibroblasts. This value was higher for the Vita CAD-Temp surface, which leads to the assumption that these cells expand better over this type of material.

To date, we are unaware of any other *in vitro* study concerning these two transepithelial abutment surfaces on which these cellular parameters have been studied. Nevertheless, of note may be the article by Atay et al., in which cytotoxicity and cellular apoptosis on various ceramic and polymeric materials produced by a CAD/CAM system are studied. This group includes IPS e.max CAD and Vita CAD-Temp, where it is shown that the latter presents low cytotoxicity levels [7].

These authors point out the lack of studies in which the biological behavior of the various dental materials available on the market is evaluated, with most articles focused primarily on their physical and chemical behavior. 

Another study comparing these materials is that by Mörmann WH et al., which compares the surface of natural enamel with that of artificial enamel, analyzing, among other parameters, the levels of roughness using SEM. This group of researchers looked for a material that resembles the natural dentition as much as possible in order to achieve a natural surface and that prevents the appearance of biofilm on the surface. They concluded that all the materials studied behaved in an almost identical way in this regard [25].

The biological properties of restoration materials are essential to obtain good results, particularly in the case of implants. Specifically, given that the intermediate abutments are structures in close contact with the implant and with gingival tissue, the cell lines of this tissue must be adapted around them [10,25,26,35]. Keratinocytes represent the main line of keratinized gingiva, whereas gingiva fibroblasts create the extracellular matrix, collagen and stroma that surround the keratinocyte layer [6,36].

As Grenade et al. pointed out fairly recently, lithium disilicate attachments were introduced, which, like those of PMMA, interact with the soft gingival tissue and influence the stabilization of this tissue around the implant crown. It is especially important that this joint be strong enough to form an airtight barrier that protects the peri-implant tissue from bacterial invasion [9]. 

Bacterial adhesion and consequent initial biofilm composition depend on the topography, hydrophobicity of the surface and existing contact between the different microorganisms. Oral biofilm is one of the most studied microbiological systems and tends to form mainly around the interfaces between one material and another [37]. 

Adaptation of this fibroblast cell line depends on factors such as the material surface, cellular phenotype or composition of the material. In this study, we decided to use human fibroblasts from an immortal cell line in order to obtain results more easily extrapolated to all types of patients. Other studies use fibroblasts extracted from the connective tissue of a patient from whom the third molar or even cell lines of mice are extracted, which have antigen limitations or other intrinsic characteristics of each species that can lead to error when extrapolating the results obtained to other cases [6,24].

Roughness is an extensively analyzed parameter in various articles in which different implant element surfaces are studied. In the case of materials used in this study, this value is lower in the VITA CAD-Temp (0.38 ± 0.02 μm), which, according to studies, also provides comfort to patients given that friction on soft tissues is reduced (Mörmann). Pendegrass et al. concluded that cell adhesion and proliferation improved in values between 0.03 and 0.3 μm, whereas Pabst obtains values <1 μm as smooth surfaces and >1 μm as rough surfaces, establishing typical roughness for dental materials above 1.3 microns [6,36]. 

Traditionally, it has been said that fibroblasts adhesion is better on smooth surfaces, although there is no current consensus regarding the ideal material surface in contact with mucosa. Quicker contact has been seen on smooth surfaces, which would coincide with the results of the study in which images were taken at 24 h. An interesting question is whether there is a need to create a gingival fibroblast barrier stable enough to prevent the passage of bacteria to the area. These are capable of producing bacterial colonies in just 8 h. Provisional material such as VITA CAD Temp provides roughness with low values, as demonstrated in the results, and producing adequate cell growth and spreading can positively influence implant integrity [6,10,24,38,39] 

This *in vitro* study has some limitations. The roughness values obtained are good for fibroblast proliferation but it could have different reaction with C. albicans adhesion and proliferation. Previous studies showed that the Ra parameter should be near 0.2 micrometers. It could be interesting to study the surface characteristics in relation to bacterial colonization in future studies [40,41,42,43].

It is true that, although the values obtained in this study may bear some similarity to those collected in other published articles regarding roughness, it should not be compared to those irregularities, since, in this case, we carried out the profilometry study in accordance with Standard ISO 25178. As commented by Fischer et al., Kournetas et al. and Rizo-Gorrita et al., it may be necessary to revisit the need to obtain roughness parameters in a different manner for *in vitro* studies, given that the Ra value is an arithmetic value that depends on experimental conditions and the technique used on the surface [10,21,39].

Another factor that influences cell adhesion is the composition of the material, which we included in Table 2, in the case of this article. We presented the results in percentage of mass for each element according to the materials analyzed.

On the one hand, IPS e.max CAD, according to the manufacturer, consists of the following compounds: SiO_2_, Li_2_O, K_2_O, MgO, Al_2_O_3_, P_2_O_5_, ZnO and other oxides. When analyzing the discs in this study, according to EDS, we could verify there was a high proportion of silicon and oxygen. All the elements indicated by the manufacturer were present with the exception of lithium, which, due to its low molecular weight, often goes unnoticed in EDS analysis. Other authors, such as Riquieri et al., have experienced a similar situation when obtaining the results of their studies [44].

Vita CAD-Temp is composed, according to the manufacturer, of PMMA or poly (methyl 2-methylpropenoate) (C_5_O_2_H_8_), silicon dioxide, pigments and MPRP (microfilled reinforced polyacrylate). In EDS analysis, we detected carbon and oxygen, as well as sulfur and silicon. As we know, carbon is the main element of PMMA chains, but we can also find other elements such as hydrogen, oxygen, nitrogen, chlorine, sulfur or fluorine. In this case, hydrogen was not detected [45]. The Vita CAD-Temp manufacturer assert there is no residual monomer in CAD/CAM polymerization and we couldn’t find it either. 

Although the SEM images of the disc surfaces show a slightly more regular and concentric surface in the case of Vita CAD-Temp, both surfaces are not homogeneous. Perhaps this little difference is one factor that favors the best expansion of fibroblasts on this surface. Nevertheless, this is the only statistically significant difference we have found between the two materials.

The influence of the material’s components in the surrounding tissues is still not understood. Some studies report that this can affect the quantity and bound protein type, as well as its conformation, orientation and binding strength [23].

Another parameter that indicates the level of cellular response is the shape of the cytoskeleton, which coordinates the regulatory signals that the fibroblast receives from the medium, in addition to helping the coordination of cell size and function. In this study, only the nuclei coating was calculated, but it is possible to calculate the percentage of cell coating, whose formula is indicated by Rizo-Gorrita et al. [10,46,47]. The formula is as follows: % nuclear area × $\frac{área\;nuclear}{área\;celular}$.

Applying this formula, the cell coating on the IPS e-max disc was $\frac{21.26}{0.306}=69.47\%,$ and it was $\frac{20.57}{0.306}=67.22\%$ on the PMMA disc.

In addition to the aforementioned parameters, there are others such as wettability or cytotoxicity that can influence early adhesion to the surface of the study materials. To analyze them all, it is necessary for future *in vitro* studies to study more variables, using standards (such as ISO 10993: 2009), so as to objectively compare results in future studies. 

## 5. Conclusions

With the results obtained in this study, we conclude that Vita CAD-Temp is a material that, although widely used for other dental applications, demonstrates a cellular behavior that is similar to those of lithium disilicate (current gold standard) and is, therefore, a material suitable for use as an implant attachment in daily practice, as it helps shape the gum around the implant crown.

More *in vitro* and clinical studies will be necessary where the benefits of this polymeric material can be proven. 

## Figures and Tables

**Figure 1 polymers-11-00744-f001:**
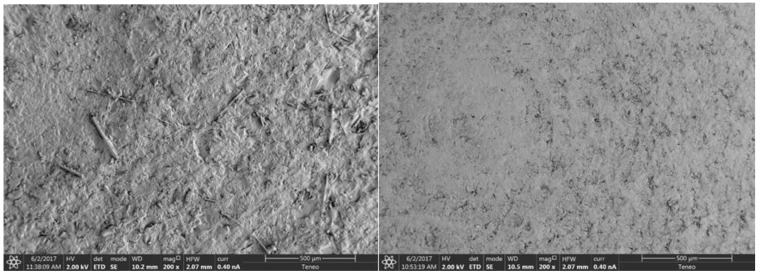
Images captured using SEM of the surfaces of IPS e.max CAD discs (**left**) and Vita CAD-Temp (**right**) at a magnification of 200×.

**Figure 2 polymers-11-00744-f002:**
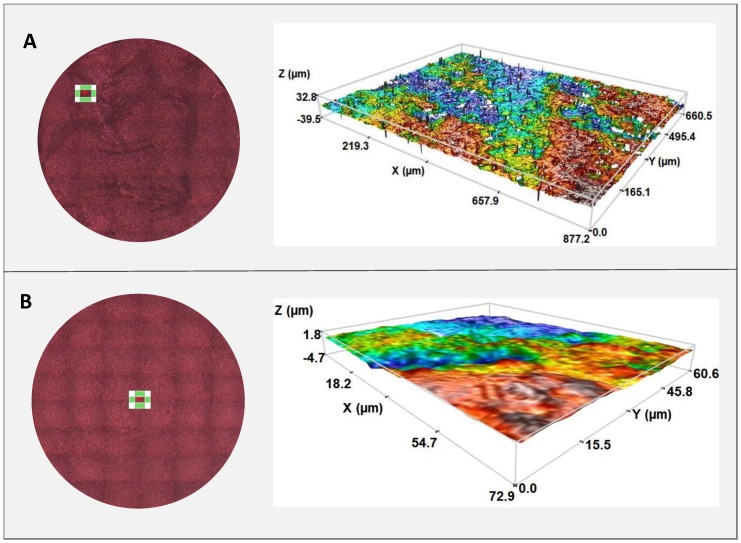
Measurement areas and 20× objective profilometry for each IPS (**A**) and VITA CAD Temp (**B**) disc.

**Figure 3 polymers-11-00744-f003:**
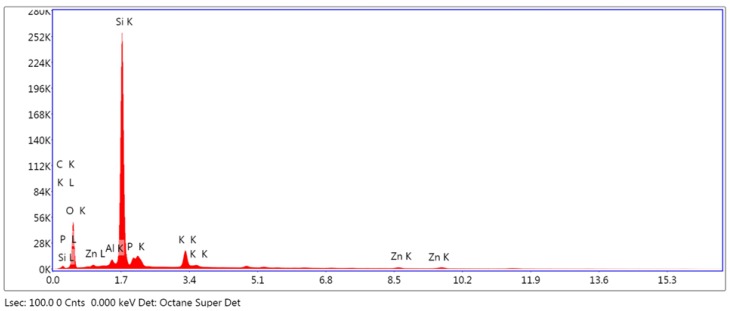
Energy dispersive spectroscopy analysis of IPS e-max CAD.

**Figure 4 polymers-11-00744-f004:**
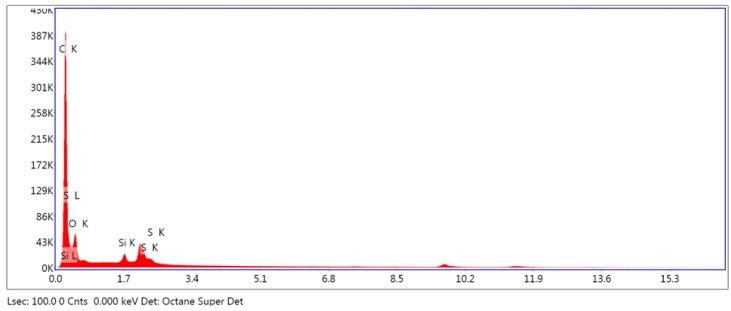
Energy dispersive spectroscopy analysis of VITA CAD-Temp.

**Figure 5 polymers-11-00744-f005:**
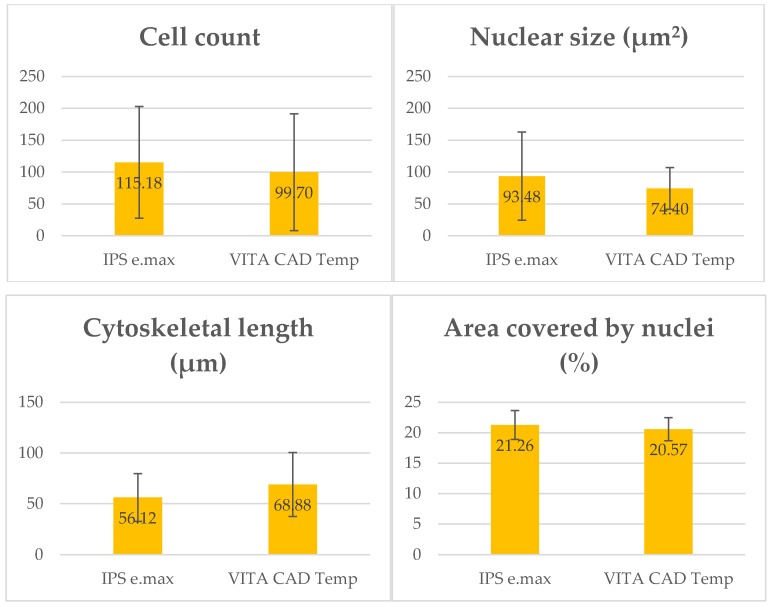
The graphs above represent the number of cells, average size, cytoskeleton length and area covered by nuclei on the two types of disc being studied. We obtained these data by averaging the values taken from the five regions of interest of each disc.

**Figure 6 polymers-11-00744-f006:**
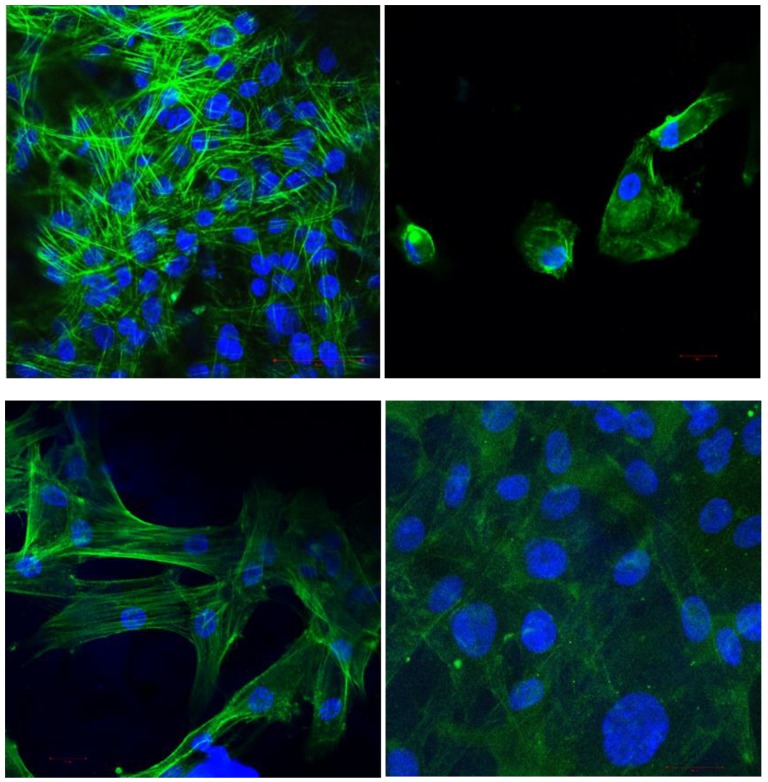
Confocal microscope images of fibroblasts on IPS at 40× and 63× (top left and right, respectively) and VITA CAD Temp (lower images).

**Table 1 polymers-11-00744-t001:** Summary indicating the name of the material brands under study, type of material, composition, manufacturer and reference and batch numbers.

Material under Study	Material Type	Composition	Manufacturer	Ref./Batch
IPS e.max CAD	Vitreous ceramic lithium disilicate	SiO_2_, Li_2_O, K_2_O, MgO, ZnO, Al_2_O_3_, P_2_O_5_	Ivoclar Vivadent, Schaan, Liechtenstein	HT A1/C 14 REF #626407 LOT V28352
Vita CAD-Temp monoColor	Polymethacrylate (PMMA)	C_5_O_2_H_8_, SiO_2_ and pigments.	VITA Zahnfabrik, Bad Säckingen, Germany.	1M2T LOT 51750

**Table 2 polymers-11-00744-t002:** Percentage of mass (wt %) broken down into elements, obtained through energy dispersive spectroscopy (EDS) analysis of IPS e.max CAD and Vita CAD-Temp monoColor samples.

Element	IPS e.max CAD(wt %)	Element	Vita CAD-Temp monoColor (%)
C	8.13	C	68.20
O	44.21	O	29.77
Al	1.20	Si	0.55
Si	36.72	S	1.49
P	4.18		
K	4.63		
Zn	0.93

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
