# Peer review of "In vitro Comparative Study of Fibroblastic Behaviour on Polymethacrylate (PMMA) and Lithium Disilicate Polymer Surfaces"

_polymers, 2019, doi:10.3390/polym11040744_

Round 1

Reviewer 1 Report

This study presents very interesting and important research for dental practitioners and researchers,

The really important issues are a creative research concept and practical soundness of the results and conclusions.

However, I think that Authors have to rewrite the introduction and focus on importance of cell proliferation on abutment surface and cell proliferation mechanism. I suggest to cancel the history of materials which is widely known and limit the description of materials.

Furthermore, the discussion starts from null hypothesis confirmation when the null hypothesis is not provide earlier. Therefore I suggest to clarify the null hypothesis at the end of introduction.          I recommend to extend the discussion about information why your study is important for practitioners and confront your results with other authors in a clinical point of view.

Finally the language of the manuscript has to be revised by native speaker.

Author Response

1.      “I think that Authors have to rewrite the introduction and focus on importance of cell proliferation on abutment surface and cell proliferation mechanism. I suggest to cancel the history of materials which is widely known and limit the description of materials.”

Thank you for your feedback. We have rewritten the Introduction, focusing on the importance of cell proliferation on the abutment surface of a cell proliferation mechanism, as you suggested (Paragraph 8, Lines 74-79). We have also deleted the history of the materials, as you were right when you said it was widely known.

2.      The discussion starts from null hypothesis confirmation when the null hypothesis is not provide earlier. Therefore I suggest to clarify the null hypothesis at the end of introduction.”

As you noticed, we forgot to write the null hypothesis at the end of the Introduction, so, following your suggestion, we added it to the last paragraph (Paragraph 10, Lines 84-87).

3.      “I recommend to extend the discussion about information why your study is important for practitioners and confront your results with other authors in a clinical point of view.”

We really appreciate your recommendation about the discussion. This is an important section of our manuscript, and we hope it is now complete (Paragraph 47-49, Lines 264-276).

4.      “the language of the manuscript has to be revised by native speaker”

We sent the manuscript to a native English editor to correct the grammar, spelling and punctuation errors and to make improvement to the style wherever necessary.

Reviewer 2 Report

Thanks for the article , but there are many flaws in the articles, such as:

The paragraphs: all should be re-written - the current version is very poorly written that one sentence one paragraph. 

Fig.5 - all units in y-axes are missing

p3L 91: baked? should be sintered right (and what are the dimension of the Lithium dilisicate after sintering) ? Also the SEM as shown in Fig 1 is a polished surface or non-polished surface? Clinically do these materials need to be polished before using at patient mouth? What is the milling bit parameter ? It seems that all these parameters are not carefully controlled.

Different materials, different roughnesses, different surface characteristics, different chemistries and so on would contribute to various effects on the fibroblasts. So, what's the most important message that you want to bring to the readers? Sadly there is not enough information that can be conclusive here.

UV sterilization - what's the operation parameters, e.g. wavelength, irridiance/power? how much time that you have worked the sterilization before the biological test?

The results part: You have mentioned the statistics in M&M but none of the results really have p values in comparison. So, what have you compared?

Author Response

1.      “The paragraphs: all should be re-written - the current version is very poorly written that one sentence one paragraph.”

Following your advice, we have sent the manuscript to a native English editor to improve the manuscript’s writing and to correct the errors wherever necessary.

2.      “Fig.5 - all units in y-axes are missing”

We extended the figure description in order to clarify the graph’s units.

3.      “p3L 91: baked? should be sintered right (and what are the dimension of the Lithium dilisicate after sintering) ? Also the SEM as shown in Fig 1 is a polished surface or non-polished surface? Clinically do these materials need to be polished before using at patient mouth? What is the milling bit parameter ? It seems that all these parameters are not carefully controlled.”

First of all, we’d like to thank you for your suggestions, which surely improved our manuscript’s quality. As you noticed, we made a spelling mistake when we wrote “baked” instead of “sintered,” but we corrected this (Paragraph 14, Line 100). In the same paragraph (Lines 100-102), we detailed the final dimension of lithium disilicate discs after the milling and sinterization process. Also, we added a new paragraph in the M&M section, where we specified the discs’ milling process (Paragraph 13, Lines 95-99). We also compared our results with clinical studies, where the writers talk about excellent end results using both materials (references [32, 33, 34]).

4.      “Different materials, different roughnesses, different surface characteristics, different chemistries and so on would contribute to various effects on the fibroblasts. So, what's the most important message that you want to bring to the readers? Sadly there is not enough information that can be conclusive here.”

Ours is a comparative in vitro study between two widely known materials in the dentistry field. We decided to compare the lithium disilicate as a gold standard not only in the crowns composition but also in the implant abutment manufacturing, with an interim material like PMMA. As other studies confirm, using provisional abutment is important for soft tissue adaptation, which improves aesthetical and health characteristics.

5.      “UV sterilization - what's the operation parameters, e.g. wavelength, irridiance/power? how much time that you have worked the sterilization before the biological test?”

The UV sterilization operation parameters, according to the manufacturer, were added in M&M (Paragraph 15, Lines 103-104). On the other hand, the sterilization procedure lasted 30 minutes, as we already specified in our manuscript (Paragraph 15, Line 104).

6.      “The results part: You have mentioned the statistics in M&M but none of the results really have p values in comparison. So, what have you compared?”

Thank you for your observance; we have added p values and their significance in the Results section (Paragraph 42-44, Lines 237-243).

Reviewer 3 Report

It is interesting work related with dental materials used in clinical practice and their properties. These type of works usually are considered not only in the field of materials science, but also (more strongly)  in the area of their practical implications. This work is focused on the chosen properties of two dental materials dedicated to CAD/CAM systems developed by one manufacturer. This limit scientific soundness of the work - we may feel unsatisfied because more similar materials from other manufacturers have not been tested which would increase the universality. However, the work is still interesting.

I have also some suggestions to the Authors.

-          First, EDS method is not suitable for chemical analyses of mass percentage of polymeric materials. The technique should not be used here as quantitative, at most as qualitative. Moreover, for the ceramic material you show elements not mentioned in the chemical composition by manufacturer (Zn) (table 1)  but other like Li, Mg,  were not detected. Please explain it (excluding Li, because you have done it). Also please explain the presence of Si i S in PMMA composition. It should be done in discussion.  Please also  note that you wrote that it is “microfilled reinforced polyacrylate” so it will be interesting to prepare suitable sample to see used microfiller and maybe analyze it.

-          You wrote:  “Another factor that influences cell adhesion is the composition of the material, which in the case 311 of this article, is included in Table 2.” – if yes, please comment you result in that contact. You didn’t do it. Without it, the value of you EDS is rally limited.

-          “The average roughness (...)” – what parameter exactly? Ra?

-          You mentioned microbiological behavior of dental materials and their relation with surface properties. Hoverer, you should also mention, that PMMA materials tends to be colonized by C. albicans, and the fact that numerous works are focused on elimination of this problem, e.g.

Acosta-Torres, L.S.; Mendieta, I.; Nuñez-Anita, R.E.; Cajero-Juárez, M.; Castaño, V.M. Cytocompatible antifungal acrylic resin containing silver nanoparticles for dentures. J. Int. J. Nanomed. 2012, 7, 4777–4786. ; Chladek, G.; Basa, K.; Mertas, A.; Pakieła, W.; Żmudzki, J.; Bobela, E.; Król, W. Effect of Storage in Distilled Water for Three Months on the Antimicrobial Properties of Poly(methyl methacrylate) Denture Base Material Doped with Inorganic Filler. Materials 20169, 328. ; Nam, K.Y.; Lee, C.H.; Lee, C.J. Antifungal and physical characteristics of modified denture base acrylic incorporated with silver nanoparticles. Gerodontology 2012, 29, 413–419.

Next you should provide short discussion about measured surface parameters, with potential colonization of PMMA be candida.  Please note that some work suggest in that context, that Ra parameter should at the level 0,2 micrometers.

Abuzar, M.A.; Bellur, S.; Duong, N.; Kim, B.B.; Lu, P.; Palfreyman, N.; Surendran, D.; Tran, V.T. Evaluating surface roughness of a polyamide denture base material in comparison with poly (methyl methacrylate). J. Oral Sci. 2010, 52, 577–581.

Similar discussion should be performed for ceramic material, especially if we notice that roughness was 4-times larger than for PMMA. So you recognize that parameter as favorable, in that context? The Candida and bacterium colonization, related to roughness can have also influence on peri-implant zone.... You results suggest, that PMMA is better, due to the much lower roughness?

-          Please give the results of statistical analyses in the Results section.

Author Response

1.      “First, EDS method is not suitable for chemical analyses of mass percentage of polymeric materials. The technique should not be used here as quantitative, at most as qualitative. Moreover, for the ceramic material you show elements not mentioned in the chemical composition by manufacturer (Zn) (table 1) but other like Li, Mg,  were not detected. Please explain it (excluding Li, because you have done it). Also please explain the presence of Si i S in PMMA composition. It should be done in discussion.  Please also note that you wrote that it is “microfilled reinforced polyacrylate” so it will be interesting to prepare suitable sample to see used microfiller and maybe analyze it.”

Thank you for your suggestion. Riquieri et al. (reference 44) analysed our material too; they also found silicon in its conformation; Astasov-Frauehoffer et al. analysed our composition material in 2018 as well (reference 45). We had forgotten to write ZnO as a chemical composition by the manufacturer (as we see in page 5 of this document https://www.emaxrevolution.com/assets/ifu-emax-press_english.pdf), but we corrected Table 1.  

We consider that EDS analysis is a good tool for better characterization of the studied materials. In the analysis, lithium and magnesium elements were measured but not detected, possibly due to their low molecular weight (Li Z=3, Mg Z=11). This issue has been reported by other authors like Ramos et al. (Ramos Nde, C.; Campos, T.M.; Paz, I.S.; Machado, J.P.; Bottino, M.A.; Cesar, P.F.; Melo, R.M. Microstructure characterization and SCG of newly engineered dental ceramics. Dent. Mater. 2016, 32, 870–878.): “(…) as expected, lithium could not be identified in any of the analyzed materials,” when comparing Vita Suprinity with lithium disilicate ceramics; or like Riquieri et al. (reference 44): “(…) lithium, one of the main elements of both materials, could not be estimated by EDS due to its low molecular weight.”

Hence, in our opinion, it is important to compare our EDS analysis results with those of the manufacturer in order to clarify the discrepancy in the presence of lithium.

Thanks to your observation, we specified the typical PMMA composition in Lines 253-258. Despite that, the manufacturer does not specify the presence of Si and S, and we know PMMA is composed of hydrogen, oxygen, nitrogen, chlorine, sulfur or fluorine, besides the carbon.

The Vita CAD-Temp “microfilled reinforced polyacrylate” analysis could be of great scientific interest. We will suggest this to other more specialized research groups because, in our opinion, it should contribute to improving the readers’ knowledge.

2.      “You wrote:  “Another factor that influences cell adhesion is the composition of the material, which in the case 311 of this article, is included in Table 2.” – if yes, please comment you result in that contact. You didn’t do it. Without it, the value of you EDS is rally limited.”

We added a comment to the first mention of the relation between the adhesion and composition of the material in Lines 364-366.

3.      “The average roughness (...)” – what parameter exactly? Ra?

We added the roughness parameter that we used (Sa) in Paragraph 38 and Line 210. It was previously explained in M&M (Lines 116-117).

4.      “You mentioned microbiological behavior of dental materials and their relation with surface properties. Hoverer, you should also mention, that PMMA materials tends to be colonized by C. albicans, and the fact that numerous works are focused on elimination of this problem, e.g.” “Next you should provide short discussion about measured surface parameters, with potential colonization of PMMA be candida.  Please note that some work suggest in that context, that Ra parameter should at the level 0,2 micrometers.” “Similar discussion should be performed for ceramic material, especially if we notice that roughness was 4-times larger than for PMMA. So you recognize that parameter as favorable, in that context? The Candida and bacterium colonization, related to roughness can have also influence on peri-implant zone.... You results suggest, that PMMA is better, due to the much lower roughness?”

We appreciate your suggestion, and we think the bacteria adhesion and colonization parameters in relation with the material surface are incredibly important for future studies. The oral cavity, as we all know, has bacteria that can colonize the de peri-implant zone, which can produce an implant failure. We focused our study on the fibroblast behaviour on both of the material surfaces. We added this information in Paragraph 63, Lines 332-335, placed on page 10.

Round 2

Reviewer 1 Report

Good job! I don't have any further suggestions.

Author Response

1.      Good job! I don't have any further suggestions.

Thank you for your indications.

Reviewer 2 Report

Thanks for resubmitting the article, still there are some problems:

VITA claim their CAD Temp is MMA free (https://www.vita-zahnfabrik.com/en/VITA-CAD-Temp-multiColor-25330,27568.html) , I believe they might have done some tesst on the residual MMA. So, logically speaking, in the polymerization there is no MMA to form PMMA in their CAD/CAM block. In your article you have cited they are PMMA-based composite - is there any evidence? Have you testified? 

P2L43: Lithium disilicate is not porcelain! Porcelain refers to one specific type of feldspathic  ceramic. Lithium disilicate is can be a type of glass ceramic or silica-based cceramic. Please amend.

First Paragraph in introduction. - You should mention what materials that dental CAD/CAM can mill. Please put 2nd paragraph and 3rd paragraph together. Paragraphs 4-6 can be put together.Paragraphs 7-9 should put together. Paragraphs 10-11 should put together. 

Your focus should be on the implant crown rather than abutment- is it a norm to put CAD-Temp as abutment? You can mention the bacterial accumulation at the margin of the crown. I believe when the materials are obtaining CE marks or some medical device registrations, biocompatibility tests such as ISO 10993 have been done. So, what's the purpose to try this? Are you simulating the clinical situation that you will try UV sterilization before placing an crown? Please try to rewrite the story.   

UVC sterilization - you should try to cite some articles to proof UVC can sterilize dental ceramics or CAD/CAM block. You have mentioned UVC 100-280nm. However, commonly we define vacuum UV as 100-200nm and UVC is 200-280nm. So, when the UVC sterilization is on, what is the peak wavelength? Also, the 4.43- 12.40 eV means the photon energy which can be directly converted from the wavelength. There is no need to mention. UV can functionalize the inorganic oxides in ceramic blocks, so it may happen the same here in particular you have used fibroblasts that might have good response in OH group. As a whole, you are evaluating the fibroblastic behaviour after UVC irridiation rather than sterilization...

Disc size: 8mm x 2 mm (what is the tolerence)? 

P10L294 , zirconium --> zirconia

Fig. 5 --> Very poor figure. All should be redrawn.

Why don't you polish the surface to certain roughness before you carry on the biological experiment? This makes the biological experiment without any standard.

Given the SD of cell counts and nuclei sizes are so big, do you think you can use other method to present your results, such as 95% CI?

Fig. 6  CAM --> CAD

P11L361 : c clbicans --> C. albicans

Author Response

1.      VITA claim their CAD Temp is MMA free (https://www.vita-zahnfabrik.com/en/VITA-CAD-Temp-multiColor-25330,27568.html) , I believe they might have done some tesst on the residual MMA. So, logically speaking, in the polymerization there is no MMA to form PMMA in their CAD/CAM block. In your article you have cited they are PMMA-based composite - is there any evidence? Have you testified? 

We agree with the manufacturer in the fact that we can’t find residual monomer after Vita CAD Temp polymerization, so we decided to specify this in our manuscript Indroduction (Line 66-67) and Discussion (Lines 363-364). This has some advantages such as the absence of contraction and mucosal irritation, CAD/CAM also improve the accuracy (Atay A, Gürdal I, Bozok Çetıntas V, Üşümez A, Cal E. Effects of New Generation All-Ceramic and Provisional Materials on Fibroblast Cells. Journal of Prosthodontics. 2018;28:e383-e394).

2.      P2L43: Lithium disilicate is not porcelain. Porcelain refers to one specific type of feldspathic ceramic. Lithium disilicate is can be a type of glass ceramic or silica-based cceramic. Please amend.

We appreciate your collaboration. We used ceramics and porcelain as a synonymous, but we have corrected in the manuscript (P2L43). 

3.      First Paragraph in introduction. - You should mention what materials that dental CAD/CAM can mill. Please put 2nd paragraph and 3rd paragraph together. Paragraphs 4-6 can be put together.Paragraphs 7-9 should put together. Paragraphs 10-11 should put together. 

We added the materials that can be milled using CAD/CAM in P1L41-42. We also linked the mentioned paragraphs.

4.      Your focus should be on the implant crown rather than abutment- is it a norm to put CAD-Temp as abutment? You can mention the bacterial accumulation at the margin of the crown. I believe when the materials are obtaining CE marks or some medical device registrations, biocompatibility tests such as ISO 10993 have been done. So, what's the purpose to try this? Are you simulating the clinical situation that you will try UV sterilization before placing a crown? Please try to rewrite the story.   

UVC sterilization - you should try to cite some articles to proof UVC can sterilize dental ceramics or CAD/CAM block. You have mentioned UVC 100-280nm. However, commonly we define vacuum UV as 100-200nm and UVC is 200-280nm. So, when the UVC sterilization is on, what is the peak wavelength? Also, the 4.43- 12.40 eV means the photon energy which can be directly converted from the wavelength. There is no need to mention. UV can functionalize the inorganic oxides in ceramic blocks, so it may happen the same here in particular you have used fibroblasts that might have good response in OH group. As a whole, you are evaluating the fibroblastic behavior after UVC irradiation rather than sterilization...

Nowadays many dental implants studies talk about emergence profile creation. In our study, we focused on IPS e. max and VITA CAD Temp as materials for implant abutments, right after the second surgery is done and fibroblasts reorganization around abutment is expected to be achieved, it was not focused on a prosthetic crown point of view.

CE marks are performed to obtain a non-carcinogenic certification or non-allergic certification, etc. but in the field of dental materials, analyzing the cellular behavior and size measure parameters are regular procedures that are not done in an ISO or CE certificate.

We have corrected the interval for UVC. Regarding UV sterilization, it is also a regular procedure for evaluating cell behavior on dental materials, similar studies can be found: Rutkunas V, Bukelskiene V, Sabaliauskas V, Balciunas E, Malinauskas M, Baltriukiene D. Assessment of human gingival fibroblast interaction with dental implant abutment materials. J Mater Sci Mater Med. 2015 Apr;26(4):169. doi: 10.1007/s10856-015-5481-8; Grenade C, De Pauw-Gillet MC, Gailly P, Vanheusden A, Mainjot A. Biocompatibility of polymer-infiltrated-ceramic-network (PICN) materials with Human Gingival Fibroblasts (HGFs). Dent Mater. 2016 Sep;32(9):1152-64. doi: 10.1016/j.dental.2016.06.020.

In any case, the reviewer is right that any procedure can influence the outcome of the experiment, which is why you should be especially careful in describing the methods.

5.      Disc size: 8mm x 2 mm (what is the tolerence)? 

Straumann® CARES® Series CAD/CAM system has this tolerance value ± 3 µm.

6.      P10L294 , zirconium --> zirconia

Thanks for the observation, we have already corrected (P9L276).

7.      Fig. 5 --> Very poor figure. All should be redrawn.

We have looked through the figure 5 and we have corrected all the errors we noticed and now we canceled the decimals in y axis. Now we think the reader could understand what we want to show.

8.      Why don't you polish the surface to certain roughness before you carry on the biological experiment? This makes the biological experiment without any standard.

When we decided to start our study we wanted to analyze both materials in a pure state. This is a limitation of our study, because we agree with you on the fact that maybe we would have obtained better results using standard roughness values. We are sure we’ll do it in future studies.

9.      Given the SD of cell counts and nuclei sizes are so big, do you think you can use other method to present your results, such as 95% CI?

We have corrected this. We have reviewed the data, and these are correct. The explanation to said SD in some cases is the biological variability, which despite standardizing the methods, cannot be eliminated 100%. The 95% CI could have been used but it is still a dispersion measure, similar to the SD.

10.   Fig. 6  CAM --> CAD

We corrected the spelling mistake in P9L270 (Fig. 6).

11.   P11L361 : c clbicans --> C. albicans

We have switched “c. albicans” to “C. albicans”.

Reviewer 3 Report

Necessary corrections have been made.

Author Response

1.      Necessary corrections have been made.

Thank you for your indications.